# Analysis of accumulated SARS-CoV-2 seroconversion in North Carolina: The COVID-19 Community Research Partnership

**John C. Williamson**[1,2]*, **Thomas F. Wierzba**[1], **Michele Santacatterina**[3], **Iqra Munawar**[1], **Austin L. Seals**[4], **Christine Ann Pittman Ballard**[1], **Martha Alexander-Miller**[5], **Michael S. Runyon**[6], **Lewis H. McCurdy**[7], **Michael A. Gibbs**[6], **Amina Ahmed**[8], **William H. Lagarde**[9], **Patrick D. Maguire**[10], **Robin King-Thiele**[11], **Terri Hamrick**[11], **Abdalla Ihmeidan**[11], **Shakira Henderson**[12], **T. Ryan Gallaher**[13], **Diane Uschner**[3], **Sharon L. Edelstein**[3], **David M. Herrington**[4ᵒ], **John W. Sanders**[1ᵒ], **on behalf of the North Carolina sites of the COVID-19 Community Research Partnership**[¶]

1 Department of Internal Medicine, Section on Infectious Diseases, Wake Forest Baptist Health, Winston-Salem, North Carolina, United States of America, 2 Department of Pharmacy, Wake Forest Baptist Health, Winston-Salem, North Carolina, United States of America, 3 Department of Biostatistics and Bioinformatics, Biostatistics Center, George Washington University, Rockville, Maryland, United States of America, 4 Department of Internal Medicine, Section on Cardiovascular Medicine, Wake Forest Baptist Health, Winston-Salem, North Carolina, United States of America, 5 Department of Microbiology and Immunology, Wake Forest School of Medicine, Winston-Salem, North Carolina, United States of America, 6 Department of Emergency Medicine, Atrium Health's Carolinas Medical Center, Charlotte, North Carolina, United States of America, 7 Department of Internal Medicine, Division of Infectious Diseases, Atrium Health, Charlotte, North Carolina, United States of America, 8 Department of Pediatrics, Atrium Health, Charlotte, North Carolina, United States of America, 9 Department of Pediatrics, WakeMed Health and Hospitals, Raleigh, North Carolina, United States of America, 10 Department of Radiation Oncology, New Hanover Regional Medical Center, Wilmington, North Carolina, United States of America, 11 Campbell University School of Osteopathic Medicine, Lillington, North Carolina, United States of America, 12 Center for Research and Grants, Vidant Health, Greenville, North Carolina, United States of America, 13 Department of Infectious Diseases, Vidant Health, Greenville, North Carolina, United States of America

ᵒ These authors contributed equally to this work.
¶ Members of the North Carolina sites of the COVID-19 Community Research Partnership are listed in the Acknowledgments.
* johnwill@wakehealth.edu

**Data Availability Statement:** All relevant data are within the manuscript and its Supporting Information files.

## Abstract

## Introduction

The COVID-19 Community Research Partnership is a population-based longitudinal syndromic and sero-surveillance study. The study includes over 17,000 participants from six healthcare systems in North Carolina who submitted over 49,000 serology results. The purpose of this study is to use these serology data to estimate the cumulative proportion of the North Carolina population that has either been infected with SARS-CoV-2 or developed a measurable humoral response to vaccination.

## Methods

Adult community residents were invited to participate in the study between April 2020 and February 2021. Demographic information was collected and daily symptom screen was

**Funding:** For the conduct of this study, institutions received funding from the CARES Act of the U.S. Department of Health and Human Services (DHHS) through the State of North Carolina (PS Award #30263, author JWS). The funding agency played no role in any aspect of this study, including study design, data collection, analysis, preparation of manuscript, and the decision to publish.

**Competing interests:** The authors have declared that no competing interests exist.

completed using a secure, HIPAA-compliant, online portal. A portion of participants were mailed kits containing a lateral flow assay to be used in-home to test for presence of anti-SARS-CoV-2 IgM or IgG antibodies. The cumulative proportion of participants who tested positive at least once during the study was estimated. A standard Cox proportional hazards model was constructed to illustrate the probability of seroconversion over time up to December 20, 2020 (before vaccines available). A separate analysis was performed to describe the influence of vaccines through February 15, 2021.

## Results

17,688 participants contributed at least one serology result. 68.7% of the population were female, and 72.2% were between 18 and 59 years of age. The average number of serology results submitted per participant was 3.0 (±1.9). By December 20, 2020, the overall probability of seropositivity in the CCRP population was 32.6%. By February 15, 2021 the probability among healthcare workers and non-healthcare workers was 83% and 49%, respectively. An inflection upward in the probability of seropositivity was demonstrated around the end of December, suggesting an influence of vaccinations, especially for healthcare workers. Among healthcare workers, those in the oldest age category (60+ years) were 38% less likely to have seroconverted by February 15, 2021.

## Conclusions

Results of this study suggest more North Carolina residents may have been infected with SARS-CoV-2 than the number of documented cases as determined by positive RNA or antigen tests. The influence of vaccinations on seropositivity among North Carolina residents is also demonstrated. Additional research is needed to fully characterize the impact of seropositivity on immunity and the ultimate course of the pandemic.

## Introduction

Estimating the proportion of the population previously infected with SARS-CoV-2, the agent of COVID-19, or who have been successfully vaccinated is imperative to optimally characterize the epidemiology of the pandemic and to make informed public health decisions about when and how to resume normal activities. Using case definitions based on clinically motivated testing for SARS-CoV-2 RNA or antigens is not reliable for multiple reasons. SARS-CoV-2 infections may not be recognized among asymptomatic or mildly symptomatic individuals [1–3]. In some communities, the lack of available testing for COVID-19 limited the ability to detect or diagnose cases, especially in the first few months of the pandemic. Often in rural areas access to care and testing is limited by external resources such as transportation. Our research group has also demonstrated that large-scale population-based cross-sectional sero-surveillance is similarly problematic because of rapid sero-reversion, especially among people with mild or asymptomatic disease [4].

To overcome these limitations, we established the COVID-19 Community Research Partnership (CCRP), a population-based longitudinal syndromic and sero-surveillance study. The CCRP includes >17,000 participants who submitted at least one serology result since April 16, 2020. These participants were recruited from six healthcare systems in North Carolina between

mid-April 2020 and February 2021. Over 49,000 longitudinal serology tests from CCRP participants were recorded, including some participants who completed up to eight sequential serology tests. The purpose of this study is to use these serology data to estimate the cumulative proportion of the population enrolled in our study that has either been infected with SARS-CoV-2 or developed a measurable humoral response to vaccination.

## Materials and methods

Only the sero-surveillance portion of the CCRP in North Carolina is described in this paper. Community residents age 18 years or older within six North Carolina health systems were invited to participate in the study using multiple methods of communication, including email, websites, health system communications, and social and mass media (radio and television). Potential participants in two of the systems, Wake Forest Baptist Health and Atrium Health, were initially invited on April 16[th], 2020. Potential participants in the other four health systems, WakeMed, New Hanover Regional Medical Center, medical associates of Campbell University School of Osteopathic Medicine, and Vidant Health were invited in November 2020. All participants provided informed consent for study procedures, including those required to secure a blood sample for serology testing. In the consent process, interested persons were provided a secure link to online informed consent. Demographic information was collected and daily symptom screen was completed using a secure, HIPAA-compliant, online portal. Participants were queried in the portal to determine healthcare worker status. The CCRP study was approved by the IRB of Wake Forest University Health Sciences.

A portion of participants were selected for serological testing. These were chosen to demographically represent the populations living in the region served by the health system. Participants were mailed kits for in-home collection of capillary blood via finger prick. The kits contained a lateral flow assay (LFA) to be used in-home to test for presence of anti-SARS-CoV-2 IgM or IgG antibodies. LFA results were recorded and interpreted using a smartphone application with central review (Scanwell Health, Inc. © 2020). In the first three months of the study, participants received a LFA by Syntron Bioresearch Inc., which detects IgM and IgG antibodies to the SARS-CoV-2 nucleocapsid antigens. However, this assay became unavailable during the study period. Beginning in July 2020, participants received the Scanwell SARS-CoV-2 IgM IgG Test from Teco Diagnostics, which detects IgM and IgG antibodies to the spike protein and nucleocapsid antigens. A subset of participants received two 20 μL volumetric absorptive microsamplers (Mitra®, Neoteryx) for sample collection, and these were analyzed centrally using the Syntron LFA. Both LFAs were validated at the Frederick National Laboratory for Cancer Research (FNLCR) by the National Cancer Institute (NCI) using a panel of antibody-positive samples from patients with PCR confirmed SARS-CoV-2 infection or pre-pandemic controls [5, 6].

Participants were mailed test kits at various times throughout the study period depending on test kit availability, supply chain disruptions, and shipping delays, all of which were generally influenced by the pandemic itself. Participant enrollment occurred in an ongoing (rolling) fashion over time so that participants who enrolled earlier in the study period had more opportunities to be tested. Likewise, the decision by some to stop participating in the study limited the number of tests that could have been performed for these individuals. Lastly, the number of tests performed for each participant was influenced by the participant's willingness to complete each test or return samples for central testing.

Accumulated SARS-CoV-2 seroconversion, testing positive for IgM and/or IgG at least once during the study period, was estimated. This is presented as the probability of prior infection from the beginning of the study up to December 20, 2020 (end of observation period), the

time when vaccines were made available to certain members in the study population. A standard Cox proportional hazards model was constructed to illustrate the probability of seroconversion over time, taking into consideration covariates of age, sex, and healthcare worker status. The proportional hazards assumption was tested and not rejected using the Schoenfeld residuals [7]. A separate analysis was performed to describe the influence of vaccines. In this analysis, the period of observation was extended to February 15, 2021, and given the high likelihood of vaccination among healthcare workers, the Cox model was stratified by healthcare worker status. Results for this period represent the probability of prior infection or vaccination. Because of the dynamic nature of the CCRP population, with some dropping out after a period of participation, the data were censored on the day after the last negative serology. Participants who reported a serology result after the last day of observation for each analysis were considered censored on the last day if all prior serologies were negative. In the analysis, the hazards of healthcare worker status and biological sex were non-proportional, violating the proportionality hazard assumption of standard Cox model. We therefore estimated average hazard ratios (AHR) by using a weighted Cox regression [8, 9] to evaluate the effect of age and sex on time to seroconversion. Similar to the standard hazard ratio, an AHR of 1 indicates no difference in survival rates across all time points. An AHR greater than 1 means an increased risk, while an AHR lower than 1 means a reduction in risk over time [10]. Separate models were fit for healthcare workers and non-healthcare workers. All analyses were performed using R version 4.0.2 [11].

## Results

Within the CCRP population of North Carolina, 17,688 participants contributed at least one serology result. Characteristics of these participants are listed in Table 1. 68.7% of the population were female and 72.2% of participants were between 18 and 59 years of age. Approximately 11% reported being a member of a minority race/ethnic group. The average number of serology test results submitted per participant was 3.0 (±1.9). Healthcare worker profession was reported for 35.2% of study participants and 79.3% were female. The average number of serology test results submitted per healthcare worker was 3.5 (±2.1), which was higher than non-healthcare workers (2.7 ±1.7).

By December 20, 2020, the overall probability of seropositivity in the CCRP population since the beginning of the study was 32.6% (95% CI 28.4, 35.0). This probability can be considered the probability of prior infection since vaccines were not available for most people before this date. Fig 1 illustrates the accumulating probability over time. Many participants in the CCRP study identified as healthcare workers and were in the initial target group to receive a COVID-19 vaccine. Table 2 lists estimates of the probability of seropositivity before and after the availability of COVID-19 vaccines and according to healthcare worker status. By February 15, 2021 the probability among healthcare workers and non-healthcare workers was 83% (82, 85) and 49% (95% CI 47, 52), respectively. The analysis at this date represents the probability of either prior infection or vaccination. Fig 2 demonstrates a clear inflection upward in the curve around the end of December, which suggests a significant impact of vaccinations on serology results, especially for healthcare workers. Prior to the inflection, the probabilities were relatively close, suggesting that healthcare workers were not becoming infected at an appreciably higher rate than non-healthcare workers. A life table (Table 3) provides cumulative probabilities of seroconverting over time which correspond to the model illustrated in Fig 2.

Table 4 lists hazard ratios for risk of seroconverting among subgroups. Prior to December 20, 2020 neither sex nor age posed a significant risk of seropositivity among non-healthcare workers. By February 15, 2021, males were 19% less likely to have seroconverted during the

**Table 1. Characteristics of North Carolina CCRP participants in the serology analysis (n = 17,688).**

|  | Number (%) |
|---|---|
| Age (years) |  |
| 18–39 | 5,049 (28.5) |
| 40–59 | 7,719 (43.6) |
| 60+ | 4,920 (27.8) |
| Sex |  |
| Female | 12,160 (68.7) |
| Male | 5,528 (31.3) |
| Race/Ethnicity |  |
| Black or African American | 542 (3.1) |
| Hispanic or Latinx | 432 (2.4) |
| Other | 1,042 (5.9) |
| White | 15,672 (88.6) |
| Healthcare Worker Status |  |
| No | 11,461 (64.8) |
| Yes | 6,227 (35.2) |
| Healthcare System Location |  |
| Atrium Health | 2,732 (15.4) |
| Campbell University | 325 (1.8) |
| New Hanover Regional | 506 (2.9) |
| Vidant Health | 649 (3.7) |
| Wake Forest Baptist Health | 11,558 (65.3) |
| WakeMed | 1,918 (10.8) |
| Vaccination Reported (after Dec 20, 2020) |  |
| Yes | 8,041 (45.5) |
| No | 9,647 (54.5) |

observation period (AHR 0.81, 95% CI 0.73, 0.90). Age did not have a significant impact on the risk of seroconverting among non-healthcare workers. As for the healthcare workers, again prior to December 20, 2020 the risk of seroconverting was not different for any of the sub-groups. However, by February 15, 2021 males were 10% more likely to have seroconverted, and older age groups were less likely to have seroconverted. The oldest age group of 60+ years was 38% less likely to have seroconverted (AHR 0.62, 95% CI 0.54, 0.72).

## Discussion

Results of the COVID-19 Community Research Partnership (CCRP) suggest there may be more infections occurring in North Carolina than is documented based on reporting of positive SARS-CoV-2 RNA or antigen tests. Using US Census estimates of the total population in NC in 2019 and the number of reported positive tests according to the NC Department of Health and Human Services (DHHS) as of March 3, 2021 [12], the cumulative incidence of COVID-19 in NC is calculated to be approximately 8.3%, a number that is significantly less than the probability of prior infection on December 20, 2020 reported here (32.6%).

There are many aspects of the CCRP that are uniquely capable of determining the likelihood of prior infection in North Carolina. Unlike other serology studies that relied on cross-sectional analysis [13–17], the CCRP assessed serology status among participants over time with multiple possible measurements per participant. This is especially important as emerging evidence has documented short-term duration of seropositivity associated with SARS-CoV-2

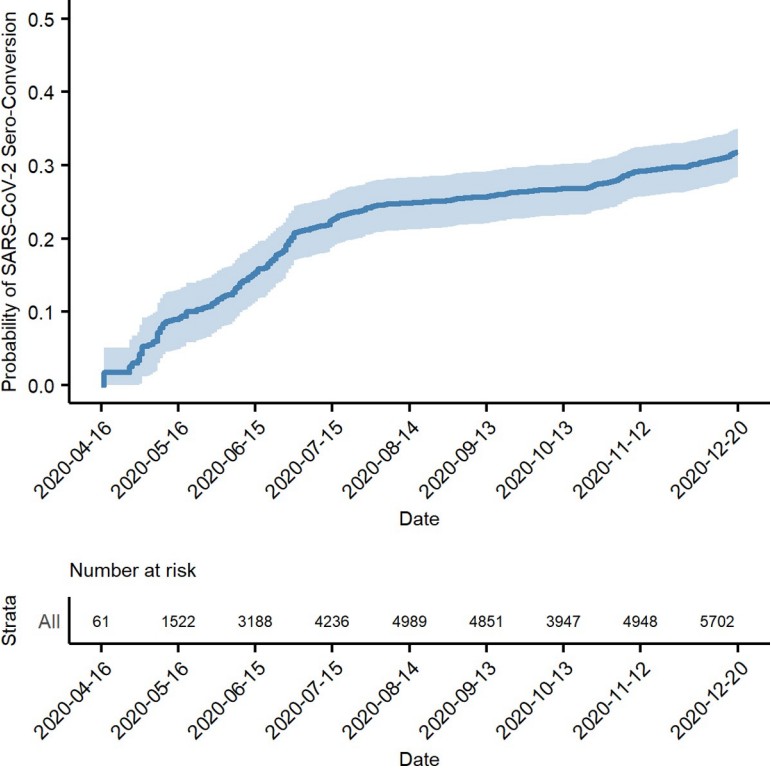

**Fig 1. Probability of prior SARS-CoV-2 infection before the availability of vaccines.**

infection, as short as 2 months duration [4]. Traditional methods of performing cross-sectional studies to identify seroprevalence would fall short and underestimate progress toward population immunity [18, 19]. Even with the advantage of multiple measurements over time, there is still some risk of underestimation within the CCRP population. Individuals who joined the CCRP study relatively late in the pandemic may have already been infected and subsequently sero-reverted before their first test.

In order to fully characterize rates of SARS-CoV-2 infection during a pandemic, it is critically important to begin the process of identifying infections as early as possible. Efforts to identify infections during the CCRP study began in April 2020, an early stage of the pandemic and well ahead of peak infections that would occur in the coming winter months. In addition, the long time span (April 2020 to February 2021) of the study allowed for more thorough capture of seroconversions in the population and therefore the ability to determine accumulation of seropositivity, including among members of the population who would not have sought

**Table 2. Probability of seropositivity according to healthcare worker status.**

| Period of Observation | Estimate (95% confidence interval) | |
|---|---|---|
| | **Non-healthcare worker** | **Healthcare worker** |
| April 16, 2020 to December 20, 2020[1] | 0.35 (0.31, 0.38) | 0.27 (0.23, 0.31) |
| April 16, 2020 to February 15, 2021[2] | 0.49 (0.47, 0.52) | 0.83 (0.82, 0.85) |

1. Probability of prior infection (before vaccines).

2. Probability of prior infection or vaccination.

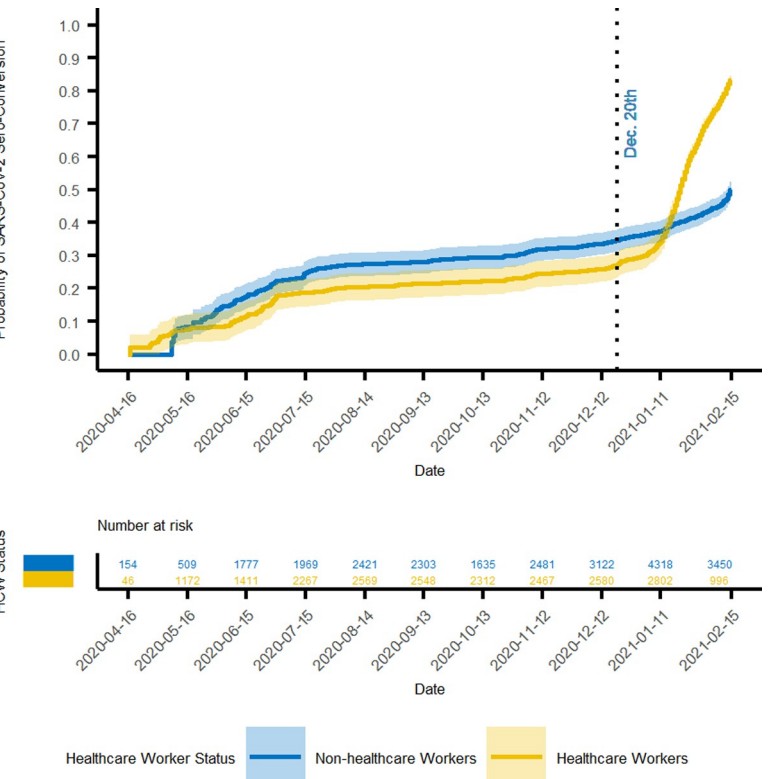

**Fig 2. Probability of prior SARS-CoV-2 infection or vaccination.**

testing, e.g. asymptomatic infections or symptomatic individuals who have a known positive contact.

A COVID-19 vaccination campaign began in December 2020 in North Carolina and initially targeted healthcare workers and people age 65 or older. Owing to the nature of the study, which solicited participation within health system networks, the proportion of CCRP

**Table 3. Cumulative probability of seroconverting over time.**

| Date | At risk | | Events | | Censored | | Probability (95% CI) | |
|---|---|---|---|---|---|---|---|---|
| | Non-HCW | HCW | Non-HCW | HCW | Non-HCW | HCW | Non-HCW | HCW |
| 4/16/2020 | 154 | 46 | 0 | 0 | 0 | 0 | 0.00 (0.00,0.00) | 0.00 (0.00,0.00) |
| 5/16/2020 | 509 | 1172 | 19 | 31 | 20 | 37 | 0.08 (0.04,0.12) | 0.07 (0.03,0.12) |
| 6/15/2020 | 1777 | 1411 | 115 | 61 | 266 | 63 | 0.17 (0.14,0.21) | 0.11 (0.07,0.16) |
| 7/15/2020 | 1969 | 2267 | 162 | 170 | 960 | 274 | 0.24 (0.21,0.28) | 0.19 (0.14,0.23) |
| 8/14/2020 | 2421 | 2569 | 89 | 54 | 207 | 105 | 0.27 (0.24,0.31) | 0.2 (0.16,0.24) |
| 9/13/2020 | 2303 | 2548 | 22 | 34 | 211 | 94 | 0.28 (0.24,0.31) | 0.22 (0.17,0.25) |
| 10/13/2020 | 1635 | 2312 | 40 | 22 | 630 | 236 | 0.29 (0.26,0.33) | 0.22 (0.18,0.26) |
| 11/12/2020 | 2481 | 2467 | 66 | 69 | 363 | 397 | 0.32 (0.28,0.35) | 0.24 (0.2,0.28) |
| 12/12/2020 | 3122 | 2580 | 69 | 52 | 247 | 210 | 0.34 (0.3,0.37) | 0.26 (0.22,0.3) |
| 1/11/2021 | 4318 | 2802 | 212 | 339 | 722 | 365 | 0.37 (0.34,0.4) | 0.34 (0.3,0.38) |
| 2/10/2021 | 3450 | 996 | 612 | 2023 | 3007 | 618 | 0.45 (0.42,0.48) | 0.77 (0.75,0.79) |

HCW = healthcare worker.

**Table 4. Risk of seroconversion within subgroups.**

| Characteristic | December 20, 2020 | | | February 15, 2021 | | |
|---|---|---|---|---|---|---|
| | AHR | 95% CI | p value | AHR | 95% CI | p value |
| Non HCW | | | | | | |
| Sex | | | | | | |
| Female | – | – | | – | – | |
| Male | 1.15 | 0.97, 1.36 | 0.1 | 0.81 | 0.73, 0.90 | <0.0001 |
| Age Group | | | | | | |
| 18–39 | – | – | | – | – | |
| 40–59 | 1.02 | 0.81, 1.28 | 0.9 | 0.99 | 0.87, 1.48 | 0.9 |
| 60+ | 1.05 | 0.83, 1.31 | 0.7 | 1.15 | 0.99, 1.32 | 0.055 |
| HCW | | | | | | |
| Sex | | | | | | |
| Female | – | – | | – | – | |
| Male | 1.04 | 0.85, 1.29 | 0.7 | 1.10 | 1.01, 1.20 | 0.037 |
| Age Group | | | | | | |
| 18–39 | – | – | | – | – | |
| 40–59 | 1.02 | 0.84, 1.23 | 0.9 | 0.82 | 0.75, 0.88 | <0.0001 |
| 60+ | 1.10 | 0.83, 1.47 | 0.5 | 0.62 | 0.54, 0.72 | <0.0001 |

HCW = healthcare worker, AHR = average hazard ratio, CI = confidence interval.

participants who are healthcare workers was generally high and may not represent the general population. As illustrated in Table 2 and Fig 2, the proportion of healthcare workers testing seropositive prior to December 20, 2020 is quite similar to that of non-healthcare workers. This suggests that an over-representation of healthcare workers in the CCRP is not contributing to a relatively high probability of prior infection as of December 20, 2020.

It is very clear though, the availability of COVID-19 vaccinations among healthcare workers had a strong influence on serology results. There was an inflection upward in the curves after December 20, 2020, which was more prominent among healthcare workers. The increase in probability among non-healthcare workers during this interval likely reflects vaccinations received by those age 65 years or older. The subgroup analysis identified age as a factor associated with lower probability of seroconversion by February 15, 2021. This finding may be due to a longer duration of time between first vaccine dose and detectable humoral response among older vaccine recipients. The pattern of increasing probability late in the study period suggests that vaccination efforts in North Carolina are contributing significantly to the proportion of the population that have developed a humoral response to one or more SAR-CoV-2 specific antigens. Enrollment in a COVID-19 vaccine clinical trial by individuals in the CCRP study is possible prior to December 20, 2020. However, a material impact on the results of this study is not expected from the very low number of such participants.

There are other limitations to this study that need to be acknowledged. The demographics of the CCRP study population do not match that of the general population in North Carolina. There was an imbalance in sex and race, with over-representation of females and Whites. There was under-representation of young adults less than age 30. Not to mention, adolescent and pediatric residents (age <18 years) were not included in this analysis of the CCRP. For these reasons, it may not be appropriate to generalize these results to all populations in North Carolina.

The performance characteristics of LFAs in the detection of anti-SARS-CoV-2 antibodies should be considered in the interpretation of these data. The sensitivity/specificity of the Syntron LFA are: IgM 93.3%/97.5%, IgG 73.3%/100%, IgM or IgG 96.7%/97.5%. The sensitivity/specificity of the Scanwell SARS-CoV-2 IgM IgG Test are: IgM 90%/100%, IgG 86.7%/100%, IgM or IgG 96.7%/100% [5, 6]. While these performance characteristics evoke some concern about the accuracy of test results, particularly negative results, the pattern of increasing probability of seropositivity over time along with the apparent influence of vaccinations provide some measure of internal validity. Not to mention, LFAs were possibly the only practical method of determining serology status for a study population of this magnitude.

Population immunity ("herd immunity") is the point at which the incidence of infection decreases once a certain amount of the population has acquired immunity. Public health experts are particularly interested in sero-surveillance data as this helps in determining the number of infections in the population, which can be used as a surrogate of immunity. Results of the CCRP study may be particularly useful for this purpose because serology status was assessed in a longitudinal way, which for SARS-CoV-2 infections has its advantages over cross-sectional serology studies for reasons already mentioned. What is not yet known, though, is whether the cumulative proportion of the population that tested seropositive accurately represents the proportion that has acquired immunity. Indeed, it is possible that immunity may wane over time in conjunction with sero-reversion and/or declining antibody titers [1, 4, 20]. This could produce a condition in which some of the previously seropositive population has relative immunity or no immunity at all. Lastly, there are uncertainties concerning the degree to which immunity from vaccines or natural infection will extend to infections caused by newer variants of SARS-CoV-2 [21–24].

Randolph and Barreiro have calculated a population immunity threshold of 67% for SARS-CoV-2 [25]. Because the assumption that seropositivity equals acquired immunity is not yet proven for SARS-CoV-2 and because of the limitations in generalizing these results broadly, it may be premature to compare cumulative probability of seropositivity in the CCRP study to a given threshold. More research is needed to determine if the decline in cases and hospitalizations in North Carolina (February and March 2021) could be attributed to population immunity that is approaching such a threshold.

Results of the CCRP study provide valuable insights about the proportion of North Carolina residents who have been infected with SARS-CoV-2. These data suggest more North Carolina residents may have been infected than the number of documented cases as determined by positive RNA or antigen tests for SARS-CoV-2. This is consistent with the understanding that mildly symptomatic or asymptomatic individuals may not seek testing. The influence of vaccinations on seropositivity among North Carolina residents is also demonstrated. Additional research is needed to fully characterize the impact of seropositivity on immunity and the ultimate course of the pandemic.

## Supporting information

**S1 Dataset.**
(XLSX)

## Acknowledgments

**North Carolina Sites of the COVID-19 Community Research Partnership**:
  **Wake Forest School of Medicine**: Mark A. Espeland, Morgana Mongraw-Chaffin, Alain Bertoni, Allison Mathews, Brian Ostasiewski, Metin Gurcan, Alexander Ivanov, Giselle

Melendez Zapata, Marlena Westcott, Mark Mistysyn, Laura Blinson, Karen Blinson, Douglas McGlasson, Donna Davis, Lynda Doomy, Perrin Henderson, Alicia Jessup, Kimberly Lane, Beverly Levine, Jessica McCanless, Sharon McDaniel, Kathryn Melius, Christine O'Neill, Angelina Pack, Ritu Rathee, Scott Rushing, Jennifer Sheets, Sandra Soots, Michele Wall, Samantha Wheeler, John White, Lisa Wilkerson, Rebekah Wilson, Kenneth Wilson, Deb Burcombe. **Biostatistics Center of George Washington University**: Greg Strylewicz, Brian Burke, Mihili Gunaratne, Meghan Turney, Matthew Bott, Shirley Qin Zhou, Helen Huiping Wu, Ashley Hogan Tjaden, Asare Buahin, Sophia Graziani, Biyas Basak, Ya Liu. **Atrium Health**: Yhenneko Taylor, Lydia Calamari, Hazel Tapp, Michael Brennan, Lindsay Munn, Keerti L. Dantuluri, Tim Hetherington, Lauren Lu, Connell Dunn, Melanie Hogg, Andrea Price, Mariana Leonidas, Laura Staton, Kennisha Spencer, Melinda Manning, Whitney Rossman, Frank Gohs, Anna Harris, Bella Gutnik, Jennifer Priem, Ryan Burns. **WakeMed Health and Hospitals**: LaMonica Daniel. **New Hanover Regional Medical Center**: Charin L. Hanlon, Lynette McFayden, Isaura Rigo, Kelli Hines, Lindsay Smith, Alexa Drilling, Monique Harris, Belinda Lissor, Maddy Eversole, Terry Herrin, Dennis Murphy, Lauren Kinney, Polly Diehl, Nicholas Abromitis, Tina St. Pierre, Judy Kennedy, Bill Heckman, Denise Evans, Julian March, Ben Whitlock, Wendy Moore. **Vidant Health**: Michael Zimmer, Danielle Oliver, Tina Dixon, Kasheta Jackson, Martha Reavis, Monica Menon, Brandon Bishop, Rachel Roeth, Mathew Johanson, Alesia Ceaser, Amada Fernandez, Carmen Williams, Jeremiah Hargett, Keeaira Boyd, Kevonna Forbes, Latasha Thomas, Markee Jenkins, Monica Coward, Derrick Clark, Omeshia Frost, Angela Darden, Lakeya Askew, Sarah Phipps, Victoria Barnes. **Campbell University**: Chika Okafor, Regina B. Bray Brown, Pinoorma Vinod, Amber Brewster, Danius Bouyi, Katrina Lamont, Kazumi Yoshinaga, A. Suman Peela, Giera Denbel, Jason Lo, Mariam Mayet-Khan, Akash Mittal, Reena Motwani, Mohamed Raafat, Evan Schultz, Aderson Joseph, Aalok Parkeh, Dhara Patel, Babar Afridi.

**External advisory council**

Helene Gayle, Chicago Community Trust; Ruth Berkelman, Emory University; Kimberly Hanson, University of Utah; Scott Zeger, Johns Hopkins University; Cavan Reilly, University of Minnesota; Kathy Edwards, Vanderbilt University.

The authors would also like to acknowledge the excellent programmatic and technical support provided by the dedicated staff at Vysnova Partners, Inc., Oracle Corporation, Scanwell Health, Inc., SneezSafe by Sneez LLC, Neoteryx, and Javara, Inc.

## Author Contributions

**Conceptualization:** John C. Williamson, Thomas F. Wierzba, Christine Ann Pittman Ballard, David M. Herrington, John W. Sanders.

**Data curation:** Sharon L. Edelstein.

**Formal analysis:** John C. Williamson, Thomas F. Wierzba, Michele Santacatterina, Iqra Munawar, Austin L. Seals, Diane Uschner, Sharon L. Edelstein, David M. Herrington, John W. Sanders.

**Funding acquisition:** Christine Ann Pittman Ballard, David M. Herrington, John W. Sanders.

**Investigation:** John C. Williamson, Thomas F. Wierzba, Martha Alexander-Miller, Michael S. Runyon, Lewis H. McCurdy, Michael A. Gibbs, Amina Ahmed, William H. Lagarde, Patrick D. Maguire, Robin King-Thiele, Terri Hamrick, Abdalla Ihmeidan, Shakira Henderson, T. Ryan Gallaher, Sharon L. Edelstein, David M. Herrington, John W. Sanders.

**Methodology:** John C. Williamson, Thomas F. Wierzba, Michele Santacatterina, Iqra Munawar, Austin L. Seals, Martha Alexander-Miller, Diane Uschner, David M. Herrington, John W. Sanders.

**Project administration:** Thomas F. Wierzba, Christine Ann Pittman Ballard, Michael S. Runyon, Lewis H. McCurdy, Michael A. Gibbs, Amina Ahmed, William H. Lagarde, Patrick D. Maguire, Robin King-Thiele, Terri Hamrick, Abdalla Ihmeidan, Shakira Henderson, T. Ryan Gallaher, Sharon L. Edelstein, David M. Herrington, John W. Sanders.

**Resources:** Christine Ann Pittman Ballard.

**Software:** Sharon L. Edelstein.

**Supervision:** John C. Williamson, Thomas F. Wierzba, Martha Alexander-Miller, Michael S. Runyon, Lewis H. McCurdy, Michael A. Gibbs, Amina Ahmed, William H. Lagarde, Patrick D. Maguire, Robin King-Thiele, Terri Hamrick, Shakira Henderson, T. Ryan Gallaher, Sharon L. Edelstein, David M. Herrington, John W. Sanders.

**Validation:** Martha Alexander-Miller.

**Writing – original draft:** John C. Williamson.

**Writing – review & editing:** John C. Williamson, Thomas F. Wierzba, Michele Santacatterina, Iqra Munawar, Austin L. Seals, Christine Ann Pittman Ballard, Martha Alexander-Miller, Michael S. Runyon, Lewis H. McCurdy, Michael A. Gibbs, Amina Ahmed, William H. Lagarde, Patrick D. Maguire, Robin King-Thiele, Terri Hamrick, Shakira Henderson, Diane Uschner, Sharon L. Edelstein, David M. Herrington, John W. Sanders.

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
