## [Decision Letter · Decision Letter 0]

15 Jun 2021

PONE-D-21-09010

Analysis of Accumulated SARS-CoV-2 Seroconversion in North Carolina: The COVID-19 Community Research Partnership

PLOS ONE

Dear Dr. Williamson,

Thank you for submitting your manuscript to PLOS ONE. After careful consideration, we feel that it has merit but does not fully meet PLOS ONE’s publication criteria as it currently stands. Therefore, we invite you to submit a revised version of the manuscript that addresses the points raised during the review process.

Please revise the manuscript according to reviewer's instruction

We look forward to receiving your revised manuscript.

Kind regards,

Prasenjit Mitra, MD, MRSB, MIScT, FLS, FACSc, FAACC

Academic Editor

PLOS ONE

Journal Requirements:

Reviewers' comments:

Reviewer's Responses to Questions

**Comments to the Author**

1. Is the manuscript technically sound, and do the data support the conclusions?

Reviewer #1: No

2. Has the statistical analysis been performed appropriately and rigorously? 

Reviewer #1: Yes

3. Have the authors made all data underlying the findings in their manuscript fully available?

Reviewer #1: Yes

4. Is the manuscript presented in an intelligible fashion and written in standard English?

Reviewer #1: Yes

5. Review Comments to the Author

Reviewer #1: Dear authors, in fact you are reporting and many months of work, although it does not add very relevant information to current knowledge. Information on group immunity is, however, always important and for that reason publication is justified.

I have no technical recommendations to make to the article, it is clean and easy to interpret. Just a little one:

Abstract- results – it is preferable present the percentagens instead in full, for example : 66.7% instead two thirds

6. PLOS authors have the option to publish the peer review history of their article (what does this mean?). If published, this will include your full peer review and any attached files.

Reviewer #1: No

---

## [Author Response · Author response to Decision Letter 0]

21 Jul 2021

A response to reviewers is included in the attached letter titled 'Response to Reviewers'. All suggested revisions were made.

---

## [Editor Report · Decision Letter 1]

15 Nov 2021

Analysis of Accumulated SARS-CoV-2 Seroconversion in North Carolina: The COVID-19 Community Research Partnership

PONE-D-21-09010R1

Dear Dr. Williamson,

We’re pleased to inform you that your manuscript has been judged scientifically suitable for publication and will be formally accepted for publication once it meets all outstanding technical requirements.

Kind regards,

Prasenjit Mitra, MD, CBiol, MRSB, MIScT, FLS, FACSc, FAACC

Academic Editor

PLOS ONE
---

## [Editor Report · Acceptance letter]

10 Mar 2022

PONE-D-21-09010R1 

Analysis of accumulated SARS-CoV-2 seroconversion in North Carolina: The COVID-19 Community Research Partnership 

Dear Dr. Williamson:

I'm pleased to inform you that your manuscript has been deemed suitable for publication in PLOS ONE. Congratulations! Your manuscript is now with our production department. 

Kind regards, 

on behalf of

Dr. Prasenjit Mitra 

Academic Editor

PLOS ONE